# CTCF Expression is Essential for Somatic Cell Viability and Protection Against Cancer

**DOI:** 10.3390/ijms19123832

**Published:** 2018-11-30

**Authors:** Charles G Bailey, Cynthia Metierre, Yue Feng, Kinsha Baidya, Galina N Filippova, Dmitri I Loukinov, Victor V Lobanenkov, Crystal Semaan, John EJ Rasko

**Affiliations:** 1Gene and Stem Cell Therapy Program Centenary Institute, The University of Sydney, Camperdown, 2050 NSW, Australia; c.bailey@centenary.org.au (C.G.B.); c.metierre@centenary.org.au (C.M.); j.feng@centenary.org.au (Y.F.); kinsha_3@hotmail.com (K.B.); c.semaan@centenary.org.au (C.S.); 2Department of Pathology, University of Washington, 98195 Seattle, USA; gfilippo@u.washington.edu; 3Laboratory of Immunogenetics, Molecular Pathology Section, National Institute of Allergy & Infectious Diseases, 20852-8152 Rockville, USA; dloukinov@niaid.nih.gov (D.I.L.); vlobanenkov@niaid.nih.gov (V.V.L.); 4Cell and Molecular Therapies, Royal Prince Alfred Hospital, Camperdown, 2050 NSW, Australia

**Keywords:** CTCF, tumour suppressor gene, haploinsufficiency, zinc finger, CRISPR/Cas9, cancer, endometrial cancer, gene editing

## Abstract

CCCTC-binding factor (CTCF) is a conserved transcription factor that performs diverse roles in transcriptional regulation and chromatin architecture. Cancer genome sequencing reveals diverse acquired mutations in *CTCF*, which we have shown functions as a tumour suppressor gene. While CTCF is essential for embryonic development, little is known of its absolute requirement in somatic cells and the consequences of *CTCF* haploinsufficiency. We examined the consequences of CTCF depletion in immortalised human and mouse cells using shRNA knockdown and CRISPR/Cas9 genome editing as well as examined the growth and development of heterozygous *Ctcf* (*Ctcf*^+/−^) mice. We also analysed the impact of *CTCF* haploinsufficiency by examining gene expression changes in *CTCF*-altered endometrial carcinoma. Knockdown and CRISPR/Cas9-mediated editing of *CTCF* reduced the cellular growth and colony-forming ability of K562 cells. CTCF knockdown also induced cell cycle arrest and a pro-survival response to apoptotic insult. However, in p53 shRNA-immortalised *Ctcf*^+/−^ MEFs we observed the opposite: increased cellular proliferation, colony formation, cell cycle progression, and decreased survival after apoptotic insult compared to wild-type MEFs. CRISPR/Cas9-mediated targeting in *Ctcf*^+/−^ MEFs revealed a predominance of in-frame microdeletions in *Ctcf* in surviving clones, however protein expression could not be ablated. Examination of *CTCF* mutations in endometrial cancers showed locus-specific alterations in gene expression due to *CTCF* haploinsufficiency, in concert with downregulation of tumour suppressor genes and upregulation of estrogen-responsive genes. Depletion of CTCF expression imparts a dramatic negative effect on normal cell function. However, CTCF haploinsufficiency can have growth-promoting effects consistent with known cancer hallmarks in the presence of additional genetic hits. Our results confirm the absolute requirement for CTCF expression in somatic cells and provide definitive evidence of *CTCF*’s role as a haploinsufficient tumour suppressor gene. *CTCF* genetic alterations in endometrial cancer indicate that gene dysregulation is a likely consequence of *CTCF* loss, contributing to, but not solely driving cancer growth.

## 1. Introduction

CTCF is a conserved multivalent transcription factor with diverse roles in transcriptional regulation and three-dimensional genome organisation such that it has been called the ‘master weaver’ protein [1]. CTCF is essential during embryonic development, as *Ctcf* null embryos are unable to implant [2]. Tissue-specific deletion of this ubiquitous factor in mice using conditional *Ctcf* alleles has highlighted the importance of CTCF availability in somatic tissues. Conditional deletion of *CTCF* in thymocytes can hamper T-cell differentiation and cell cycle progression, but not ablate T cell function [3]. Conditional deletion of *Ctcf* in the limb mesenchyme induces extensive apoptosis during limb development highlighting CTCF’s pro-survival role [4]. Similarly, deletion of *Ctcf* specifically during early mouse brain development, led to PUMA upregulation and subsequent massive apoptosis [5]. Of relevance for our studies, *Ctcf* heterozygous mice, however, are more prone to the formation of spontaneous cancers, as well those induced by radiation and chemical means [6].

CTCF links gene regulation to genomic architecture by co-ordinating DNA looping in concert with cohesin [7,8,9]. Within chromosomal territories, CTCF defines boundaries between sub-megabase-scale topologically-associated domains (TADs) [10,11,12] in a framework that is conserved [13]. These TADs themselves can serve as large gene regulatory domains establishing specific gene expression profiles [14]. TAD organisation is CTCF site orientation-specific [13,15] and rewiring of CTCF sites can significantly perturb gene expression by affecting promoter-enhancer interactions or boundaries between euchromatin and heterochromatin [16,17,18]. In cancer, hypermethylation or somatic mutation of CTCF binding sites has been shown to affect chromatin boundaries. This, in turn, can induce tumour suppressor silencing [19,20]; disruption of CTCF-dependent insulation leading to aberrant TAD formation and oncogene activation [21]; and cis-activation of genes implicated in cancer [22,23]. 

Our previous studies first demonstrated the growth inhibitory effects of CTCF in vitro [24] and subsequently confirmed that CTCF acts as a tumour suppressor gene in vivo by suppressing tumour growth [25]. Isolated *CTCF* mutations have been identified in breast, prostate and Wilms’ tumours [26] and acute lymphoblastic leukaemia [27]. However recent cancer genome studies have revealed the extensive somatic mutations occurring in *CTCF* [28]. *CTCF* has been classified as a significantly mutated gene owing to its high frequency of mutation and deletion in endometrial cancer [29]. *CTCF* mutations are detected in 35% of endometrial carcinomas exhibiting microsatellite instability (MSI), and in 20% of MSI-negative tumours [30]. One report describing 17 oncogenic signatures in cancer, defines one signature, M5, as comprising MSI-positive endometrioid cancers and some luminal A breast cancers. In this subset of endometrioid and breast cancers, *CTCF* mutations were identified in 40% of samples including inactivation of specific zinc fingers (ZFs) of CTCF that would lead to altered DNA binding [31]. We since revealed that *CTCF* genetic alterations have a pro-tumourigenic effect in endometrial cancer by altering cellular polarity and enhancing cell survival [32].

Genetic lesions in *CTCF*, whether heterozygous deletion, nonsense, frameshift, or even missense ZF mutations, can all result in *CTCF* haploinsufficiency. In endometrial cancer, *CTCF* mRNA transcripts expressed from alleles containing nonsense or frameshift mutations are subjected to nonsense-mediated decay [30,32]. Somatic missense mutations in residues critical for CTCF ZF binding to DNA can result in selective loss of binding to some CTCF target sites, but not all [26], indicating the functional implications of incomplete loss of CTCF binding in cancer is unclear. Loss of heterozygosity (LOH) at 16q22 can lead to *CTCF* haploinsufficiency and *IGF2* up-regulation in Wilms’ tumours [33]. To date, modelling the full impact of *CTCF* haploinsufficiency on CTCF’s tumour suppressor function has not been previously examined.

In this study we assessed several genetic models of *CTCF* haploinsufficiency to reveal in detail the impact of heterozygous loss of *CTCF* in somatic cells, whole mice and human endometrial cancer. Depletion of CTCF expression in K562 erythroleukaemia cells using shRNA knockdown or CRISPR/Cas9-mediated targeting of *CTCF* decreased cellular proliferation. In vivo, *Ctcf* heterozygosity negatively impacted the growth and gross development of mice. However, p53 shRNA-immortalised *Ctcf*^+/−^ mouse embryonic fibroblasts (MEFs) were functionally distinct from wild-type (WT) MEFs by exhibiting increased cellular growth and other known cancer hallmarks. Importantly, we were unable to generate *Ctcf* nullizygous MEFs after CRISPR/Cas9 genome editing confirming that CTCF is absolutely essential for somatic cell viability. Finally, we examined curated human endometrial carcinoma genomic data and observed that *CTCF* haploinsufficiency contributed to the transcriptional dysregulation of specific loci as well as inducing a unique gene signature in human cancers.

## 2. Results

We used shRNA knockdown to model the cellular consequences of reduced CTCF expression in K562 cells. Western blots showed that CTCF protein expression was significantly knocked down by ~80% in the presence of doxycycline (dox) in sh.CTCF K562 cells compared to non dox-treated cells and sh.control cells (Figure 1Ai,ii). Cellular proliferation showed that CTCF knockdown resulted in a significant reduction of proliferation (*p* < 0.0001, Figure 1B). We similarly observed a significant reduction in sh.CTCF K562 colony number compared to non dox treatment and to sh.control (both *p* < 0.0001, Figure 1C). CTCF knockdown led to growth arrest with an increase in G1 phase (*p* < 0.0001), and a concomitant reduction of cells in S (*p* < 0.0001) and G2/M phases (*p* = 0.0036, Figure 1D). We next examined the response of sh.CTCF K562 cells after UV insult and observed CTCF knockdown in K562 cells resulted in an increase in cell viability after recovery from UV exposure (*p* = 0.004) and a decrease in apoptotic (Annexin V-positive) cells (*p* = 0.0006, Figure 1E). 

These data and our previous studies indicated that CTCF dosage is critical for its tumour suppressive functions [25,32], however *CTCF* haploinsufficiency has not been definitively modelled in vitro. To address this, we used CRISPR/Cas9-directed genome editing to induce genetic lesions in K562 cells (Appendix A), which we previously verified to contain wild type *CTCF* alleles using Sanger sequencing [25]. SgRNAs were designed to direct Cas9 nuclease cleavage on both strands of the critical third exon of *CTCF*, encoding the entire N-terminus of CTCF (Appendix A). All three *CTCF* sgRNA achieved efficient Cas9 cleavage of *CTCF* exon 3 (Figure 2A). A representative Western blot showing CTCF protein expression in clones isolated after CRISPR/Cas9-mediated targeting of *CTCF* using two sgRNAs (#3 and #5) is shown in Figure 2B. CTCF protein expression was decreased by approximately 50% in most surviving clones irrespective of the sgRNA used (Figure 2C). As each clone should contain at least one edited *CTCF* allele, we PCR-amplified the edited region in *CTCF*, cloned the PCR products and then sequenced them. We detected a mixture of *CTCF* alleles arising in clones including frameshifts induced by deletion or insertions near the protospacer adjacent motif (PAM) site or in-frame deletions leading to microdeletions in the CTCF protein (Figure 2D). In some clones, we observed three distinct edited *CTCF* alleles, consistent with K562 cells having a hypotriploid karyotype [34]. SgRNAs #2 and #3 induced 100% and ~96% gene editing efficiency respectively with a ~50:50 mixture of frameshifts and in-frame deletions (Figure 2D). 

We next performed MTT cell proliferation assays on eGFP^+^ mCherry^+^ K562 cell pools (Appendix A) and showed that cells targeted using CTCF sgRNAs #2 and #3 exhibited reduced cellular proliferation (*p* = 0.014 and *p* = 0.012 respectively, Figure 2E). We also performed clonogenicity assays and confirmed that CRISPR/Cas9-directed genome editing of *CTCF* inhibited colony-forming ability by ~30 % for sgRNAs #2 and #3 (Figure 2F). Therefore, inducing genetic lesions in *CTCF* leading to haploinsufficient levels of CTCF in K562 cells had a negative impact on cellular growth. 

We then examined *Ctcf* heterozygous mice to better determine what impact heterozygous deletion of the *Ctcf* locus has on post-natal growth and development. We backcrossed these *Ctcf*^+/pgkneo^ mice (Figure 3A), which were originally described on a mixed 129SvJ:C57Bl/6J background and exhibited embryonic lethality as homozygotes [2], onto C57Bl/6J mice for at least 10 generations. Backcrossed C57Bl/6J *Ctcf*^+/pgkneo^ mice bred with wild type C57Bl/6J (WT) mice had smaller mean litter sizes than from normal WT x WT mice (Figure 3B). This was explained by both female and male *Ctcf*^+/pgkneo^ mice being born at sub-Mendelian ratios (~28% and ~24% respectively) compared to WT (*Ctcf*^+/+^) mice (both *p* < 0.0001 *Chi*-square test; Figure 3C,D). After weaning at approximately day 21 we recorded mouse weights 3 times a week until 12 weeks of age. Female *Ctcf*^+/pgkneo^ mice were smaller than WT littermates up to 7 weeks of age (~14% less body weight, Figure 3E), whereas male *Ctcf*^+/pgkneo^ mice were consistently smaller than WT littermates in the first 12 weeks of age (~12% less body weight, Figure 3F). This reduced weight phenotype was maintained in male *Ctcf*^+/pgkneo^ mice even beyond two years of age (Figure 3F). These data show *Ctcf* haploinsufficiency can significantly impact growth and development in mice. Examination of genetic variation occurring in CTCF in humans using the ExAC database [35] revealed that CTCF is extremely intolerant to genetic variation within the protein-coding region. CTCF exhibits significantly fewer nonsynonymous variants than expected (z score = 4.86) and can be classified as haploinsufficient due to intolerance to heterozygous loss-of-function variation (pLI score = 1.0; Figure 3G). Two genome-wide CRISPR screens in diploid cells [36,37] and a synthetic lethal gene trap screen in haploid cells [38] identified 916 core fitness genes essential for cell viability common to all 3 screens, including *CTCF* (Figure 3H). These data confirm CTCF as an essential gene in higher order eukaryotes.

To examine the cellular consequences of *Ctcf* haploinsufficiency, we isolated mouse embryonic fibroblasts (MEFs) from a single litter containing 4 *Ctcf*^+/pgkneo^ and 3 WT pups. These MEFs were immortalised with a retrovirus encoding stable shRNA knockdown of *p53* and then analysed by immunoblot for Ctcf protein expression (Figure 4Ai). Densitometric analysis of the *Ctcf*^+/pgkneo^ and WT MEF samples confirmed Ctcf protein was reduced in *Ctcf*^+/pgkneo^ MEFs to a mean of 58% of WT (*p* = 0.033, Figure 4Aii). We performed MTT assays and showed immortalised *Ctcf*^+/pgkneo^ MEFs exhibited an increase in cellular proliferation compared to WT MEFs (*p* = 0.0028 day 2, *p* <0.0001 day 3, Figure 4B). *Ctcf*^+/pgkneo^ MEFs also displayed an increase in colony-forming ability compared to WT (*p* < 0.0001, Figure 4C). We analysed cell cycle kinetics and showed that *Ctcf*^+/pgkneo^ MEFs exhibited a decrease in G1 phase compared to WT MEFs (*p* = 0.0072) with a concomitant increase in G2/M phase (*p* = 0.0043) (Figure 4D). We next examined the cellular response to UV-induced apoptosis and found that immortalised *Ctcf*^+/pgkneo^ MEFs exhibited a decrease in viability and concomitant increase in Annexin V-positive cells compared to WT cells (*p* = 0.0002 and *p* = 0.0005 respectively) (Figure 4E). Immortalised *p53*-deficient *Ctcf* heterozygous MEFs exhibit pro-tumourigenic characteristics, indicating that *Ctcf* is acting as a haploinsufficient tumour suppressor gene.

We next used CRISPR/Cas9-mediated targeting of *Ctcf* in monoallelic hemizygous *Ctcf*^+/pgkneo^ MEFs to assess the impact of inducing potentially deleterious *Ctcf* genetic lesions. Four sgRNAs were designed to target the first coding exon (exon 3) of *Ctcf*, as well as a control sgRNA targeting the *Rosa26* locus (Appendix A). Efficient Cas9-directed cleavage of *Ctcf* exon 3 was observed using each sgRNA against *Ctcf*, whereas Rosa26 sgRNA had no detectable effect (Figure 5A). A clonogenicity assay was performed using *Ctcf*-targeted eGFP^+^mCherry^+^
*Ctcf*^+/pgkneo^ MEFs which showed that colony-forming capacity was significantly reduced to ~30–40% of control (Figure 5B). We isolated individual clones for each *Ctcf* sgRNA by FACS and then examined Ctcf protein expression. Ctcf expression in surviving clones was maintained despite attempts to inactivate the hemizygous *Ctcf* allele, however, in some clones lower molecular weight Ctcf species were detected, e.g., for sgRNA#4 (Figure 5C), sgRNA #1, #2, #3 (Appendix A). These most likely result from in-frame deletions; or alternatively, frameshift mutations that occur in the first coding exon of *Ctcf*, such that alternative ATG start codons are utilised leading to N-terminal Ctcf protein truncations. Sequencing of CRISPR/Cas9 genome-edited surviving clones showed 44 out of 45 clones exhibited in-frame deletions or frameshift-inducing indels (Figure 5D). More than two-thirds of clones had in-frame deletions proximal to the PAM site leading to N-terminal microdeletions in Ctcf ranging in size from 1–62 aa (sgRNA#4; Figure 5E), some of which could be detected by Western blot (e.g., clones 4.2.3, 4.2.10, and 4.2.11; Figure 5C). These data confirm that CTCF is essential in somatic cells and that *CTCF* nullizygosity cannot be sustained in viable cells.

To ascertain the impact of *CTCF* haploinsufficiency in the context of human cancers, we examined a uterine corpus endometrial carcinoma (UCEC) dataset from The Cancer Genome Atlas, which exhibits *CTCF* genetic alterations in 45 out of 232 patient samples (~19%) [28]. GISTIC analysis of this cohort assigned each patient sample into potential somatic copy number alterations based on relative CTCF expression level (Figure 6A). CTCF expression was decreased in a substantial proportion of endometrial cancer samples, some of which can be directly attributed to genetic deletion of *CTCF* (deep deletion). Many inactivating nonsense and frameshift mutations in *CTCF* were found in the notionally diploid population (40 out of 179, 22.3%; Figure 6A). Samples with inactivating mutations and confirmed deletions are classified herein as ‘*CTCF*-altered’. We then analysed other gene mutations that co-occurred with or were mutually exclusive in *CTCF*-altered endometrial cancers. *TP53* mutations (66 out of 68) occurred with mutually exclusivity to *CTCF* mutations (*p* = 9.28 × 10^−6^, Figure 6B,C); whereas mutations in *MED13L*, encoding a subunit of the Mediator transcriptional co-activation complex, co-occurred with *CTCF* mutations in 13 out of 23 cancers (*p* = 2.64 × 10^−5^, Figure 6B). 

We analysed RNAseq data available for these endometrial cancer samples and showed that *CTCF* gene expression was not significantly decreased in *CTCF*-altered cancers despite the presence of inactivating mutations (Appendix A). We next examined the chromosomal distribution of all expressed genes detected above threshold in endometrial cancers (~13,000) and found an enrichment for genes expressed on chromosomes 11, 16, 22 and X (Figure 6D). However, in *CTCF*-altered cancers there was enrichment for genes expressed on chromosomes 1, 7, 9, 17, and 20 (Figure 6C). Further analysis of enriched chromosomal regions with altered gene expression highlighted multiple loci on the short arm of chromosome 17 including 17p13.1 (which contains the *TP53* locus), and the long arm of chromosome 20 (Figure 6D). *TP53* gene expression was significantly decreased in *CTCF*-altered cancers (*p* = 0.0437, Appendix A), however, there was no significant difference after the exclusion of samples containing *TP53* mutations from the analysis (Appendix A). To gain further insight, gene ontology analysis of biological processes in *CTCF*-altered endometrial cancers indicated *CTCF* mutation may impact predominantly on transcriptional regulation, cell signalling pathways, such as p53, and DNA methylation (Figure 6E). Closer examination of genes that were dysregulated in *CTCF*-altered cancers showed expression of the *CTCF* paralog *CTCFL* was decreased (*p* = 0.0167; Appendix A), the exemplar CTCF-regulated gene *H19* was decreased (*p* = 0.0087; Appendix A), whilst no change was observed in *ZFHX3* expression which is located adjacent to *CTCF* on chromosome 16q22 (Appendix A). Importantly, expression levels of the tumour suppressor genes *CDKN2A* and *PIK3CA,* which are deleted or mutated in endometrial cancer [28,39], were decreased in *CTCF*-altered samples (*p* = 0.0006 and *p* = 0.0007, respectively; Figure 6G, Appendix A). Putative tumour suppressor genes *CDH6* and *IGF2BP2* were two of the most significantly fold-decreased genes (*p* = 0.0003 and *p* = 5.71 × 10^−5^ respectively; Figure 6F, Appendix A). Furthermore, the expression of estrogen-responsive genes *KIAA1324, MLPH, MSX2, SPDEF*, *TFF3*, and *PIGR* were all significantly up-regulated in *CTCF*-altered endometrial cancers (*p* = 0.0004, 0.0007, 0.0032, 0.0009, 0.0122, and 0.0028 respectively, Figure 6F, Appendix AK–P). Lastly, differentially expressed genes in *CTCF*-altered cancers were significantly overrepresented in a 19 gene signature that classifies endometrial cancers into endometrioid and serous subtypes (8 out of 19, *p* = 1.46 × 10^−6^) [40] as well as a 320 gene classifier that distinguishes endometrioid and serous endometrial cancers from uterine carcinosarcomas (94 out of 320, *p* = 2.96 × 10^−44^) [39].

## 3. Discussion

Haploinsufficiency arises when only a single functional copy of a gene is inadequate for normal cell function [41]. *CTCF* was identified as one of nearly 300 haploinsufficient genes in humans based on published literature or a clear association with genetic disease [42]. Herein we empirically demonstrate that CTCF can be classified as haploinsufficient due to its intolerance to loss-of-function polymorphisms in humans. CTCF haploinsufficiency resulting from germline or de novo genetic mutations in *CTCF* (including genetic deletion, frameshift mutations or missense mutation) causes intellectual disability in humans; now classified as autosomal dominant mental retardation (MRD21; OMIM #615502) [43,44,45]. The impact of CTCF mutations on human gene expression manifested as a predominant downregulation of genes involved in the cellular response to extracellular stimuli [43] and hypermethylation of CTCF-binding sites [45]. The mechanisms that connect *CTCF* haploinsufficiency with cancer have yet to be elucidated.

Numerous studies over more than a decade using siRNA or shRNA knockdown of CTCF have only been suggestive of CTCF’s essential role in normal cell function. Typically, such experiments are short term, do not fully ablate CTCF expression and cells remain viable. Similarly, our shRNA knockdown of CTCF in K562 cells showed that cell proliferation and clonogenic capacity was decreased, cell survival after UV insult was increased, and cells underwent growth arrest. Paradoxically, after CTCF knockdown here, and in corroboration of our previous CTCF overexpression study [25] we observed tumour suppressive phenotypes for CTCF in K562 cells. These data reveal that physiological CTCF expression levels are critical for normal cellular function and reveal the functional importance of maintaining physiological CTCF expression levels. 

More recently CRISPR/Cas9 mediated genome editing techniques have given tremendous insight into CTCF’s roles in higher-order chromatin organisation. Most studies have focussed on rewiring CTCF-mediated chromatin interactions such as disrupting TAD boundaries [16] and switching the orientation of CTCF target sequences to alter genome topology [17]. Unbiased genetic screens using CRISPR/Cas9 sgRNA libraries or synthetic lethality in haploid cells have identified CTCF as part of an ‘essentialome’ containing ~900 core fitness genes required for cell viability [36,37,38]. However, only two studies to date have directly focused on targeting CTCF in mammalian cells using CRISPR/Cas9 editing [46,47], but with disparate outcomes. *CTCF* heterozygous MCF10A clones generated by CRISPR exhibited similar proliferation rates compared to control, though cells were slower to repair double-stranded DNA breaks [46]. Silencing of CTCF in the RS4;11 acute lymphoblastic leukaemia cell line increased colony numbers in soft agar overlays compared to control, though CTCF protein appeared to be reduced to minimal levels [47]. 

Our strategy was to examine the essentiality of CTCF in somatic cells by CRISPR/Cas9 targeting of *CTCF* in K562 cells and *Ctcf* hemizygous MEFs. It was clear that despite efficient editing of *CTCF* alleles in hypotriploid K562 cells, haploinsufficient levels of CTCF protein expression were still maintained in surviving clones. Edited cells exhibited reduced cell proliferation and colony-forming ability consistent with our shRNA knockdown of CTCF in K562 cells. However, in *Ctcf* hemizygous MEFs that only contain one coding allele of *Ctcf*, we were unable to completely abolish Ctcf protein expression using CRISPR. Accordingly, in surviving clones we detected a high incidence of in-frame microdeletions in *Ctcf*. As these microdeletions occur in the intrinsically disordered N-terminus we do not expect them to significantly impact Ctcf function. Induced frameshift deletions were also likely to produce truncated CTCF proteins initiating from alternate in-frame ATG start codons within the N-terminus. These results confirm that CTCF is absolutely required for somatic cell viability and that CTCF cannot be completely inactivated in cells. Interestingly, residual CTCF protein levels can be depleted to minimal amounts in the cell before viability is significantly impacted. Targeted degradation of CTCF in mouse embryonic stem (ES) cells using an auxin-inducible degron system highlighted that endogenous CTCF protein levels could be decreased by up to 99% for at least 2 d duration without a significant impact on cell proliferation or viability [48]. Acute depletion impacted on CTCF looping and insulation of TAD regions, but genomic compartmentalisation was maintained [48]. *CTCF*-null ES cells can progress to the blastocyst stage (E3.5) purely via retention of maternal *CTCF* mRNA, but exhibit peri-implantation lethality by E4.5–E5.5 [2].

We have also shown that CTCF has a dose-dependent impact on embryonic development, as even haploinsufficient levels of Ctcf protein affected embryonic development in mice. We observed heterozygous *Ctcf* mice being born at sub-Mendelian rates (24–28%) compared to WT littermates, which was previously suggested by a study using mice with a conditionally targeted *Ctcf* allele, but was not thoroughly quantified or statistically verified [3]. Interestingly, the same *Ctcf* hemizygous mice are born at normal Mendelian rates on a mixed C57Bl/6J:129SvJ background [6], indicating that the strain background is important factor to consider in any *Ctcf* genetic deficiency studies in mice. *Ctcf* heterozygosity also impaired normal mouse weight gain during adult development for up to seven weeks, and which then remained constant in aged male mice. As these mice were fed a normal chow diet, we could not determine whether *Ctcf* heterozygosity impairs body weight control, metabolism or nutrient signaling pathways. In future studies we will examine glucose and insulin levels in plasma after feeding-fasting cycles, body tissue composition using dual energy X-ray absorption as well as studying the impact of different chow compositions on *Ctcf* heterozygous mouse development.

*CTCF* hemizygous mice are more susceptible than WT mice to spontaneous cancer development, as well as radiation- and chemically-induced cancers [6]. Tumours in *Ctcf*^+/−^ mice compared to WT mice also exhibit increased aggressiveness in terms of invasion, metastatic dissemination and mixed epithelial/mesenchymal differentiation, confirming CTCF as a haploinsufficient tumour suppressor [6]. Our current findings showing an increase in cell proliferation, colony forming ability and numbers of cycling cells in p53-shRNA immortalised *Ctcf*^+/−^ MEFs support these conclusions. Furthermore, CTCF depletion can increase genomic instability by hindering homologous recombination repair of DNA double-stranded breaks and cause hypersensitivity to DNA damage [46]. As a result, our observation of an increase in DNA damage after UV treatment of *Ctcf*^+/−^ MEFs is consistent with impaired DNA repair. These data may explain why *CTCF* haploinsufficient MEFs in the context of additional genetic hits to p53, exhibited a number of cancer hallmarks. 

The TCGA UCEC cohort consisting of low- and high-grade endometrioid carcinomas and serous tumours were genetically defined into four categories [28]. The majority of *CTCF* somatic mutations occurred in POLE ultramutated, MSI hypermutated and copy-number low cancers, whilst copy-number high cancers with a serous-like pathology harboured *TP53* mutations [28]. This was consistent with our analysis showing mutually exclusivity between *CTCF* and *TP53* mutations in endometrial cancer. *CTCF* haploinsufficiency due to *CTCF* copy loss results in poorer survival outcomes in patients with endometrioid UCEC [6], as well as serous UCEC [32]. Our analyses provide insight into the molecular pathophysiology underlying these observations. Since CTCF is known to co-ordinate higher-order chromatin architecture to facilitate interactions between transcription regulatory sequences, our data reinforces the impact that *CTCF* haploinsufficient loss imparts in endometrial cancer via transcriptional regulation. *CTCF* haploinsufficiency results in differential regulation of genes located at specific loci, particularly on chromosomes 17 and 20, including cytoband 17p13.1 containing the *TP53* locus. Whilst this is not reflected in a significant change in *TP53* mRNA expression once accounting for the *TP53* mutation status of patient samples, genes involved in p53-mediated signal transduction are impacted. Genes involved in DNA methylation were also differentially regulated in *CTCF*-altered endometrial cancers. Molecular genetic analysis of *Ctcf*^+/−^ mice showed DNA methylation instability compared to wild type mice [6]. Divergent CpG methylation due to *Ctcf* hemizygosity was restricted to specific loci with regions within a 2 kb window surrounding divergent CpGs exhibiting a generalised pattern of DNA hypermethylation [6]. In humans with heterozygous *CTCF* mutations exhibiting an intellectual disability, specific CTCF sites exhibited DNA hypermethylation [45]. This epigenetic dysregulation may offer an explanation as to why differential gene expression was observed at particular chromosomal loci in *CTCF*-altered endometrial cancers. 

One possible hallmark of *CTCF*-altered endometrial cancers is the downregulation of tumour suppressor genes including *PIK3CA*, *CDKN2A*, *CDH6* and *IGF2BP2*. The tumour suppressor *PIK3CA* is ranked fifth after *CTCF* in the most frequently mutated genes in UCEC [6] whilst *CDKN2A* is downregulated in POLE, MSI, and copy-number low cancers compared to high-copy number cancers [28]. CDH6, which helps maintain epithelial integrity in the endometrium [49], has been shown to be a putative tumour suppressor in cholangiocarcinoma [50]. *IGF2BP2*, which was the most down-regulated gene in *CTCF*-altered endometrial cancer, was identified as a candidate tumour suppressor gene in a pan-cancer screen for homozygously deleted genes [51]. Loss of IGF2BP2 staining, which is a feature of endometrioid cancers, but not serous cancers, has been proposed as a biomarker for distinguishing endometrial tumour pathology [52].

A second hallmark of *CTCF*-altered endometrial cancers is the upregulation of estrogen-responsive genes, which includes *KIAA1324*, *MLPH*, *MSX2*, *SPDEF*, *TFF3* and *PIGR*. *CTCF* mutations do not occur in a tumour type-specific manner, but rather they define a subset of hormone-responsive cancers [31]. CTCF is a negative regulator of the pioneer factor FOXA1, which facilitates estrogen receptor interactions with chromatin in response to estrogen [53,54]. Therefore, in *CTCF* haploinsufficient endometrial tumours, FOXA1/ER interactions with chromatin may increase leading to upregulation of estrogen-responsive genes. KIAA1324, which is a positive regulator of the autophagy pathway and may protect cells from cell death, was the most upregulated gene in *CTCF*-altered endometrial cancer [55]. KIAA1324 is a marker of grade I endometrial cancer which decreases with increase in tumour grade and disease stage [56] and is a key member of gene signatures classifying histological subtypes [39,40]. Other estrogen-responsive genes upregulated in *CTCF*-altered cancers, namely *SPDEF*, *TFF3* and *PIGR*, are also components of these gene signatures, indicating that loss of *CTCF* could be an important factor determining endometrial cancer progression and pathology.

## 4. Materials and Methods

### 4.1. Cell Lines

Human erythroid leukaemia (K562) cells were grown in RPM1 1640 medium while human embryonic kidney (HEK293T) and mouse embryonic fibroblast cells were cultured in DMEM. Basal media were supplemented with 10% FCS (*v*/*v*), penicillin (100 U/mL) and streptomycin (100 μg/mL). All human cell lines were authenticated by short tandem repeat profiling (Cellbank, Westmead, Australia). 

### 4.2. Expression Vectors and Antibodies

CTCF shRNA knockdown was performed using the pFH1-UTG-CTCFshRNA lentivector and the corresponding control shRNA vector expressing *Arabidopsis thaliana* mir-159a [32]. This lentiviral vector contains eGFP and a doxycycline-inducible shRNA. For CRISPR/Cas9 genome editing: plasmid 52628-Bsd-T2A-H2B-mCherry was used to express single guide RNAs (sgRNAs) as a lentivector and was a kind gift from Yifei Liu (Yale Fertility Centre, New Haven, CT, USA). Sense and antisense oligonucleotides encoding the sgRNAs (Appendix A) were phosphorylated with T4 polynucleotide kinase (New England Biolabs, Ipswich, MA, USA), annealed and then cloned into 52628-Bsd-T2A-H2B-mCherry following *BspMI* digestion. Plasmid 53190-pLV-hUbC-Cas9-T2A-eGFP used for stable expression of a human-codon optimised Cas9 nuclease [57] was obtained from Addgene (Watertown, MA, USA). For immortalisation of MEFs, pMSCVp53.1224, a retroviral vector encoding a p53 shRNA, a kind gift from Ross Dickins (Walter and Eliza Hall Institute, Melbourne, Australia), was used. Primary antibodies include: rabbit polyclonal antibody against CTCF (1:1000) [24], mouse monoclonal antibodies against CTCF (1:1000) [58], α-tubulin (1:5000; sc-23948, Santa Cruz, Dallas, TX, USA) and GAPDH (1:5000; ab8245, Abcam, Cambridge, MA, USA). Secondary antibodies include: rabbit or mouse antibodies conjugated to horseradish peroxidase (Abcam, Cambridge, MA, USA; 1:5000).

### 4.3. Retroviral and Lentiviral Transduction

Viral supernatants were produced by calcium phosphate transfection of HEK293T cells: with pJK3, pCMVTat and pL-VSV-G packaging plasmids used to produce retroviruses; and pRSV-Rev, pMDLg/p.rre and pMD2.VSV-G used to package lentiviruses. Viral supernatants collected after 24–48 h were 0.45 μM-filtered and snap-frozen or concentrated by ultracentrifugation for 2 h at 20,000 rpm in a SW28 Beckman rotor. Viral supernatant was resuspended on ice in 10% (*v*/*v*) FCS/DMEM at 1/100 th of the original volume. Adherent cells (1–5 × 10^5^/well) were seeded in six-well plates before addition of fresh medium containing viral supernatant (~5 × 10^5^ transducing units) and Polybrene (4 μg/mL; Sigma, Zwijndrecht, The Netherlands) and ‘spin-oculated’ for 90 min at 1500 rpm. The supernatant was replaced with medium 12 h post-transduction and fluorescent cells were purified 24 h later by fluorescence activated cell sorting (FACS; >95% purity on re-analysis) using a FACS Influx (Becton Dickinson, BD, Mountain View, CA, USA). K562 cells (~5 × 10^5^/mL) in 1 mL medium with 4 μg/mL Polybrene were placed in a 5 mL capped FACS tube and transduced with viral supernatant for 90 min by ‘spin-oculation’. The cells were resuspended and incubated at 37 °C for 4 h before removal of viral supernatant. For in vitro assays, cells were either plated out immediately or allowed to recover after sorting for 48–72 h in medium containing 100 μg/mL Normocin (InvivoGen, Toulouse, France).

### 4.4. CRISPR/Cas9 Genome Editing, Validation and Molecular Genetic Analysis

SgRNAs targeting the first coding exon of human *CTCF* and mouse *Ctcf* (exon 3) were designed using the Zhang lab CRISPR design tool (crispr.mit.edu). SgRNAs targeting the adeno-associated virus integration site 1 (AAVS1) and the *Rosa26* locus were used as negative control guides in human and mouse cells respectively. We used lentiviral vectors to co-express a sgRNA with mCherry, as well as a 3XFLAG-tagged Cas9 nuclease 2A-peptide linked to eGFP. Transduced cells were FACS-enriched for eGFP^+^mCherry^+^ cells after 48 h from which gDNA was extracted from pools after 6 d for a T7 Endonuclease I assay to detect Cas9-directed DNA cleavage. We also isolated single eGFP^+^mCherry^+^ cells by FACS into 96-well plates and expanded them before isolation of genomic DNA and whole cell lysates. Genomic DNA was isolated using the Purelink Genomic DNA Extraction kit (ThermoFisher) and PCR primers were used to amplify across the targeted region (see Appendix A). PCR amplicons were denatured and re-annealed to allow heteroduplex formation, then digested with T7 Endonuclease I (New England Biolabs, Ipswich, MA, USA) according to manufacturer’s instructions and then resolved using DNA gel electrophoresis. We PCR-amplified *CTCF* exon 3 from genomic DNA isolated from K562 and *Ctcf*^+/−^ MEF clones, which had been subjected to CRISPR/Cas9-mediated gene editing, using Platinum Taq (Thermo Fisher Scientific, Waltham, MA, USA). Amplicons were ligated into pGEM-T-Easy (Promega, Madison, WI, USA) and then transformed into *E. coli*. Each clonal amplicon was then confirmed using Sanger sequencing in both directions.

### 4.5. Isolation of Mouse Embryonic Fibroblasts

All animal experiments were performed in accordance with an approved institutional animal ethics protocol from the Royal Prince Alfred Hospital Animal Welfare Committee (SSWAHS #2013/046 approved 5 August 2013). *Ctcf*^+/−^ mice were obtained on a mixed C57Bl/6:129SvJ background from the Fred Hutchinson Cancer Research Centre (Seattle, WA, USA) [2]. These mice have had the complete coding region of one *Ctcf* allele replaced with a loxP-flanked cassette containing a *pgk* promoter and *neo* gene, designated *Ctcf*^+/pgkneo^ (Figure 3A). Mice homozygous for this allele (*Ctcf*
^pgkneo^/^pgkneo^) exhibit embryonic lethality prior to embryo implantation [2]. Mice were backcrossed at least 10 generations onto C57Bl/6 mice from the Animal Resources Centre (Perth, WA, USA) before beginning phenotyping studies. Timed matings were performed with *Ctcf*^+/pgkneo^ male mice and C57Bl/6 females and female mice were checked daily for vaginal plugs. At 13.5 dpc, pregnant females were euthanised by CO_2_ asphyxiation. The uterine horns were removed and the foetuses released whilst immersed in PBS. Each pup was removed from its amniotic sac, decapitated and fetal liver removed. The carcasses were minced with a scalpel and then incubated in trypsin/EDTA solution (Invitrogen, Basel, Switzerland). The tissue fragments were triturated to break up clumps, and then concentrated using centrifugation to remove trypsin. Fresh trypsin was added to create a homogeneous solution of cellular material. The trypsin was inactivated in excess DMEM medium containing 10% (*v*/*v*) FCS and then the centrifugation step repeated. The MEFs were plated in 15 cm plates and allowed to grow for 2–3 d until there were sufficient adherent cells for cryopreservation. The remaining MEFs were transduced with MSCVp53.1224 retroviral supernatant for immortalisation. Cells were selected in 50 μg/mL hygromycin (Roche Life Science, Mannheim, Germany) to enrich for immortalised MEFs and then frozen down after 1–2 passages (P2-P3 MEFs). *Ctcf*^+/pgkneo^ mice and MEFs were genotyped according to primers listed in Appendix A. 

### 4.6. Western Blot Analysis

Protein extracts were prepared with cell lysis buffer containing 20 mM Tris-HCl (pH 8), 150 mM NaCl, 1% (*v*/*v*) Triton X-100, 0.1% (*v*/*v*) SDS, 0.5% (*w*/*v*) sodium deoxycholate, and EDTA-free protease inhibitor cocktail (cOmplete, Roche Life Science, Mannheim, Germany), prior to separation using denaturing sodium dodecyl sulfate polyacrylamide gel electrophoresis (SDS-PAGE). Proteins were transferred onto PVDF membranes in a semi-dry transfer apparatus before immunoblotting. Membranes were blocked in PBS/0.1% (*v*/*v*) Tween 20 containing 20% (*v*/*v*) BlokHen (AvesLab, Portland, OR, USA) or PBST containing 0.3% (*w*/*v*) BSA, 1% (*w*/*v*) polyvinylpyrrolidone and 1% (*v*/*v*) PEG (mw 3350). Protein expression was detected using primary antibodies followed by washing and staining with appropriate secondary antibodies conjugated to horseradish peroxidase (HRP). The HRP substrate SuperSignal^®^ Chemiluminescent Substrate (Pierce) was detected on a Kodak Imagestation 4000R Pro (Woodbridge, CT, USA) or BioRad Chemidoc Touch (Hercules, CA, USA). Blots were stripped with ReBlot Plus (Merck Millipore, Guyancourt, France) prior to re-probing with protein loading control antibodies. Densitometric analysis of bands from three independent blots was performed using ImageJ (National Institutes of Health, University of Wisconsin, USA).

### 4.7. Cell Assays

Cell proliferation was assessed by 3-(4,5-dimethylthiazol-2-yl)-2,5-diphenyltetrazolium bromide (MTT) assay (Merck Millipore, Guyancourt, France). Adherent cells (1000/well) or suspension cells (5000/well) were plated in triplicate in a 96-well plate and proliferation was assessed over 4 d by the addition of MTT at 37 °C overnight. The reaction was quenched with isopropanol/HCl and absorbance measured at 572 nm using a Polarstar Omega plate reader (BMG Labtech, Durham, NC, USA). The clonogenic capacity of adherent cells was measured by plating 1000 cells/10 cm plate in triplicate and incubating for 8–10 d. Cells were washed with PBS, fixed with ice-cold methanol and stained with Giemsa Stain (Sigma, Zwijndrecht, The Netherlands) diluted 1:20 in triple-distilled water before scoring. The clonogenic capacity of K562 cells was measured by plating 5000 cells diluted in Iscove’s Modified Dulbecco Medium (Life Technologies, Rockville, MD, USA) containing 3 mL Methocult GF H4230 (Stem Cell Technologies, Vancouver, BC, Canada) onto 35 mm gridded plates in triplicate and incubating for 8–10 d. To assess UV-induced apoptosis, cells (1 × 10^5^/well in a 12-well plate) were plated in triplicate. The following day, medium was removed from attached cells and replaced with PBS. Plates with lids removed were placed in a Stratalinker UV Crosslinker (Stratagene, La Jolla, CA, USA) and exposed to UVC irradiation (2000 μJ for MEFs, 4000 μJ for K562 cells) and allowed to recover for 18 h. Cells were harvested and stained with anti-Annexin V-APC (BD Biosciences, San Jose, CA, USA) according to the manufacturer’s protocol and with propidium iodide (PI) solution (5 μg/mL, Sigma, Zwijndrecht, The Netherlands). Cells were analysed on a Fortessa flow cytometer (BD Biosciences, San Jose, CA, USA) with analysis performed using FlowJo 9.7.6 software (Treestar, Ashland, OR, USA). Cell viability was measured after addition of PI and then analysed by flow cytometry. The viable population represents the Annexin V^-^PI^-^ cells; the apoptotic population represents the Annexin V^+^PI^−^ and Annexin V^+^PI^+^ cells combined. 

### 4.8. Cell Cycle Analysis

For cell cycle analysis by DNA content, cells were washed with PBS prior to fixation in ice-cold 70% ethanol and stored overnight at 4 °C. Post-fixation, cells were washed twice with PBS to remove all traces of ethanol prior to staining with a solution containing PI (20 μg/mL), 0.1% (*v*/*v*) Triton X-100 and RNase-A (200 μg/mL). Cells were incubated for 15 min in the dark at room temperature and analysed on a Canto-II flow cytometer (BD Biosciences, San Jose, CA, USA). Cell cycle analysis was undertaken using FlowJo 9.7.6 cell cycle modelling software (Treestar, Ashland, OR, USA) by applying the Dean-Jett-Fox algorithm. 

### 4.9. Bioinformatics Analysis

Gene expression and somatic mutation data from the uterine corpus endometrial carcinoma dataset [28] was downloaded from cBioPortal. Of the 500 samples described, 240 contain matched sequencing and copy number alteration data. Statistical tests already conducted on these data were also downloaded, including Student’s test (*p*) and Benjamini-Hochberg adjusted *p*-values (*q*). *CTCF*-altered cancers included those with somatic mutations (missense, nonsense, frameshift) and deep deletions (*n* = 45). Normal *CTCF* included shallow deletions and non-mutant samples (*n* = 178). The *CTCF*-altered gene signature used in subsequent analysis includes all differentially expressed genes (*q* < 0.05).

## 5. Conclusions

We examined CTCF essentiality and haploinsufficiency in somatic cells and mice using various molecular genetic techniques and models. Despite achieving efficient genome editing of *CTCF* using CRISPR the inability to obtain complete ablation of CTCF expression reinforces its requirement. In all cases, cellular fitness in CTCF-targeted cells was compromised leading to surviving cells compensating with reduced CTCF protein expression or truncated CTCF protein variants. Consequently cell proliferation, colony-forming ability and cycling cells were reduced. However, in the presence of additional genetic hits, such as in p53, *CTCF* haploinsufficient cells exhibited known cancer hallmarks, namely increased proliferation and reduced cell cycle control. In human endometrial cancer datasets, we identified a unique gene signature in *CTCF* haploinsufficient cancers arising from differential gene expression at specific loci. Downregulation of tumour suppressor genes and upregulation of estrogen-responsive genes may be a molecular feature of *CTCF*-altered endometrial cancers. Our study clearly demonstrates that CTCF is a haploinsufficient tumour suppressor gene that is essential for somatic cell viability and protects against cancer. As the master of weaver of the genome CTCF plays an essential role in chromatin organisation, the full impact of *CTCF* haploinsufficiency on three-dimensional chromatin architecture remains to be elucidated.

## Figures and Tables

**Figure 1 ijms-19-03832-f001:**
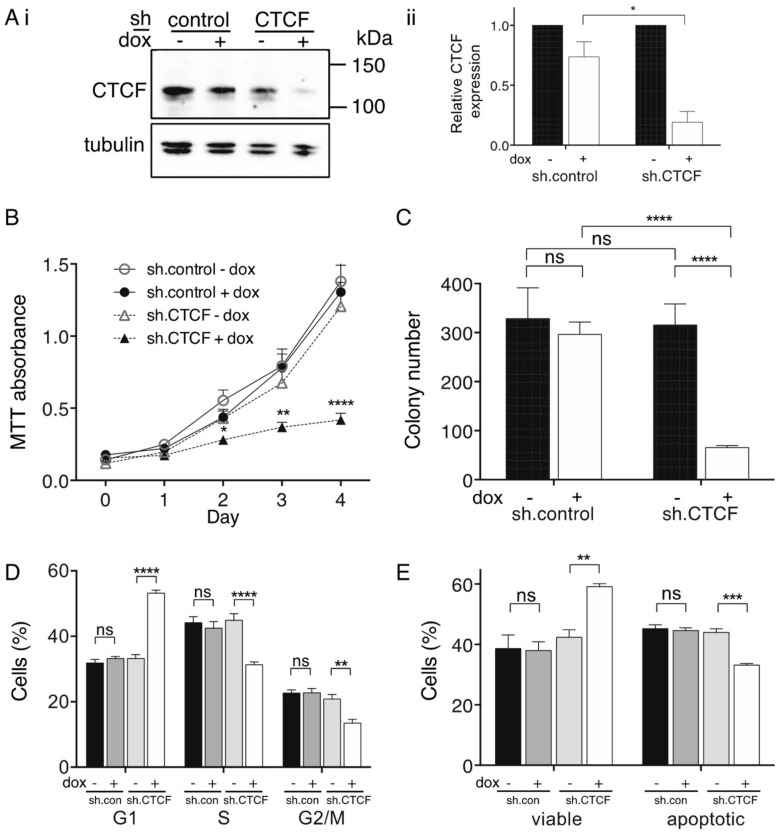
Inhibition of cell proliferation and clonogenicity following CTCF shRNA knockdown in K562 cells. (**A**) Immunoblot of CTCF shRNA knockdown in the presence and absence of doxycycline (dox) compared to control shRNA after 4 d: representative immunoblot (**i**); and relative CTCF expression normalised to α-tubulin confirmed by ImageJ densitometric analysis (**ii**). Functional assays performed after 4 d knockdown including: (**B**) MTT proliferation; (**C**) colony forming assay; (**D**) cell cycle analysis; and, (**E**) apoptotic response after recovery from UV insult. Data represent the mean ± SEM for 3 experiments each performed in triplicate. Statistical analysis was performed using a Mann-Whitney U-test (ns = not significant, * *p* < 0.05, ** *p* < 0.01, *** *p* < 0.001, **** *p* < 0.0001).

**Figure 2 ijms-19-03832-f002:**
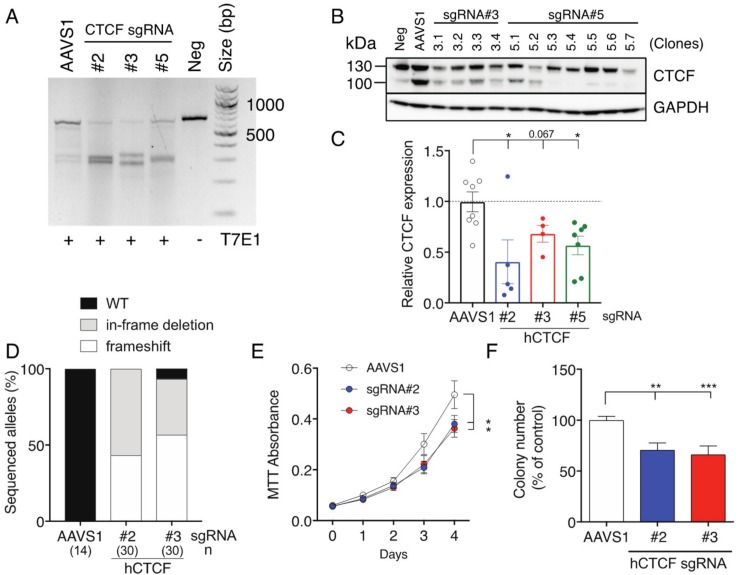
Inhibition of cell proliferation and clonogenicity following CRISPR/Cas9 targeting of *CTCF* in K562 cells. K562 cells were transduced with Cas9 and sgRNA-containing lentivectors (AAVS1 sgRNA = control; human *CTCF* exon 3 sgRNAs #2, #3, #5) and enriched for eGFP^+^mCherry^+^ cells using FACS; Neg = untransduced K562 cells. (**A**) *CTCF* exon 3 PCR amplification and T7 endonuclease I (T7EI) digestion: approximate expected sizes (in bp) for digested products #2 (310, 345), #3 (296, 359) and #5 (323, 332). Analysis of CTCF protein levels in K562 clones: (**B**) immunoblot; and (**C**) densitometric analysis of upper 130 kDa band. CTCF protein expression normalised to GAPDH expression in each sample is shown relative to untransduced K562 cells. (**D**) Summary of results after sequencing of *CTCF* exon 3 PCR amplicons from individual clones; *n* = number of clones sequences (in brackets). Functional assays performed were MTT cell proliferation (**E**); and clonogenicity assays (**F**). Quantitative data represent the mean ± SEM for 3–4 experiments each performed in triplicate. Statistical analysis was performed using a Mann-Whitney U-test (ns = not significant, * *p* < 0.05, ** *p* <0.01, *** *p* < 0.001).

**Figure 3 ijms-19-03832-f003:**
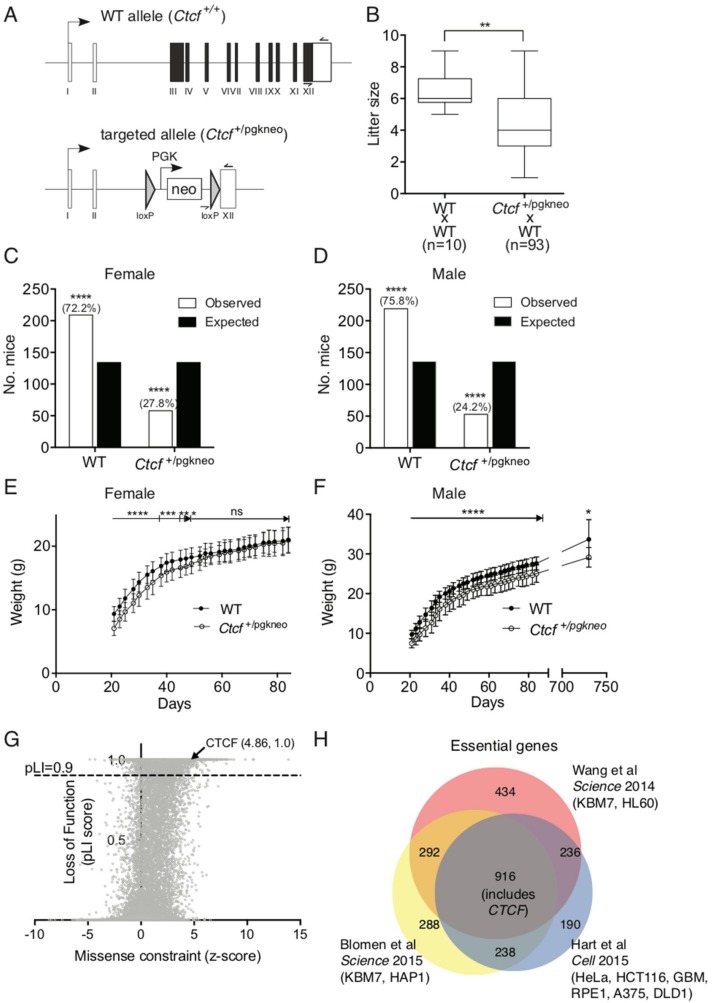
CTCF haploinsufficiency phenotype in mice and humans. (**A**) Schematic of targeted inactivation of *Ctcf* in mice. Open boxes represent untranslated regions, filled boxes represent coding region. Genotyping primers used to distinguish alleles are indicated with half-arrowheads. Litter sizes (**B**), and Mendelian ratios (%, in brackets) of female (**C**) and male (**D**) pups born from WT x *Ctcf*^+/pgkneo^ intercrosses. Weights of pups during development following weaning at day 21 (mean ± SD): (**E**) for female; and, (**F**) aged male (>2 yo) mice. (**G**) Analysis of *CTCF* genetic variation in humans using the ExAC database with pLI > 0.9 indicating intolerance to heterozygous loss-of-function variation. The missense constraint is a measure of the deviation away from the observed variants in a gene versus the expected variants (high positive z-scores indicated intolerance to variation). (**H**) Venn diagram of essential genes identified in three independent genetic screens in human cells. Statistical analysis was performed using Fisher’s exact test or Mann-Whitney U-test (ns = not significant, * *p* < 0.05, ** *p* < 0.01, *** *p* < 0.001, **** *p* < 0.0001).

**Figure 4 ijms-19-03832-f004:**
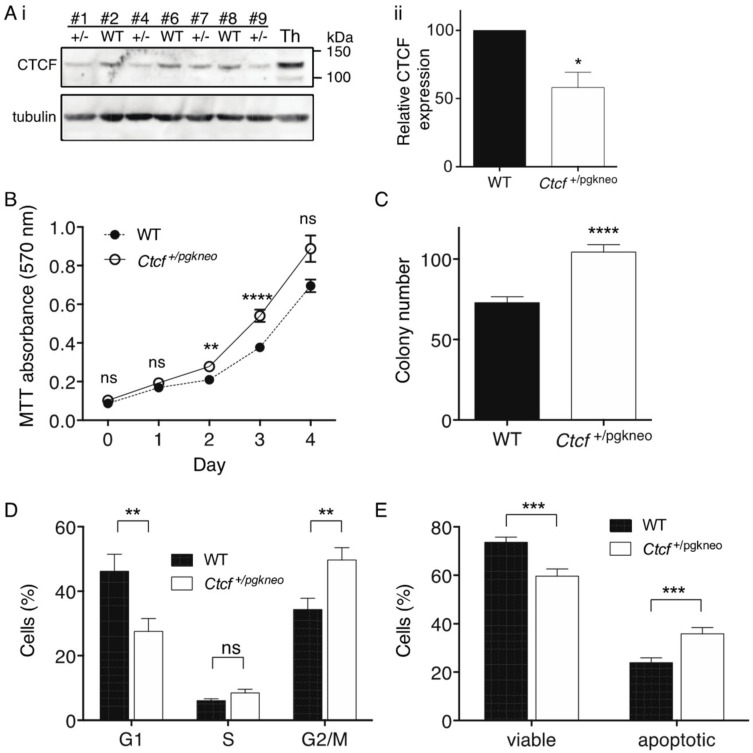
Functional characterisation of immortalised *Ctcf*
^+/−^ MEFs. (**A**) Immunoblot of whole cell lysates isolated from WT and *Ctcf*^+/pgkneo^ (+/−) MEFs (clone number indicated), Th=thymus (**i**); densitometric analysis of Ctcf protein normalised to the β-tubulin loading control (**ii**). Functional assays including: MTT proliferation (**B**); clonogenicity (**C**) cell cycle analysis (**D**); and apoptosis assay following 18 h recovery from UV irradiation (**E**). Data represent the mean ± SEM for 3 experiments each performed with 4 *Ctcf*^+/pgkneo^ and 3 WT cell lines. Statistical analysis was performed using a Mann-Whitney U-test (ns = not significant, * *p* < 0.05, ** *p* < 0.01, *** *p* < 0.001, **** *p* < 0.0001).

**Figure 5 ijms-19-03832-f005:**
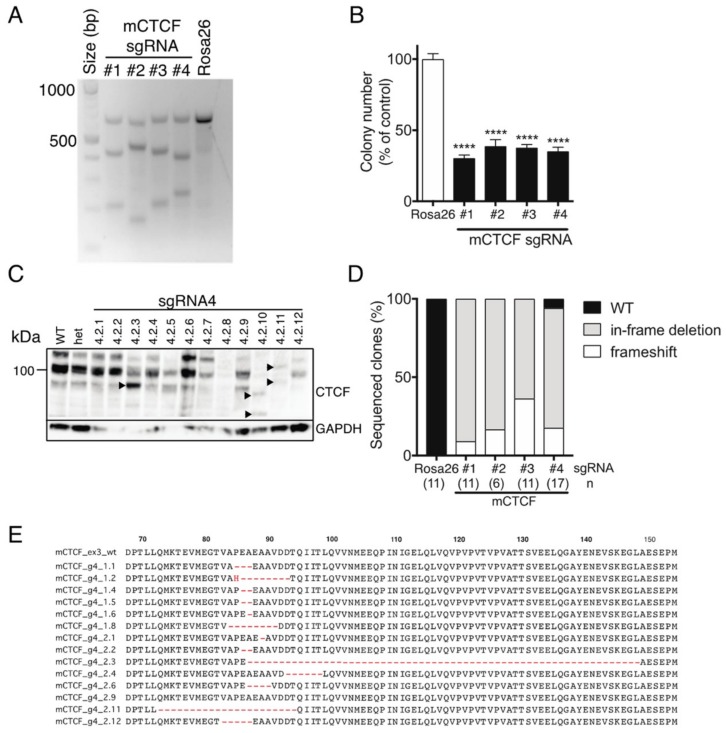
CRISPR/Cas9-directed editing of *Ctcf* in hemizygous MEFs. (**A**) *Ctcf*^+/pgkneo^ MEFs transduced with Cas9 and sgRNA-containing lentivectors (mouse *Ctcf* exon 3 sgRNAs #1, #2, #3, and #4; *Rosa26* sgRNA) were FACS-enriched and subjected to T7EI digestion of *Ctcf* exon 3 amplicons amplified from isolated gDNA. Approximate expected sizes (in bp) for digested products #1 (427, 214), #2 (476, 165), #3 (428, 213), and #4 (399, 242). Clonogenicity assay (**B**); Western blot analysis of individual clones (from sgRNA#4, arrowheads indicate lower molecular weight Ctcf variants) (**C**); and molecular genetic analysis of individual clones; n=number of clones sequenced (in brackets) (**D**). (**E**) Examples of frequently occurring in-frame deletions in *Ctcf*^+/pgkneo^ MEFs (from sgRNA#4). Quantitative data represent the mean ± SEM for three experiments each performed in triplicate. Statistical analysis was performed using Mann-Whitney U-test (ns = not significant, **** *p* < 0.0001).

**Figure 6 ijms-19-03832-f006:**
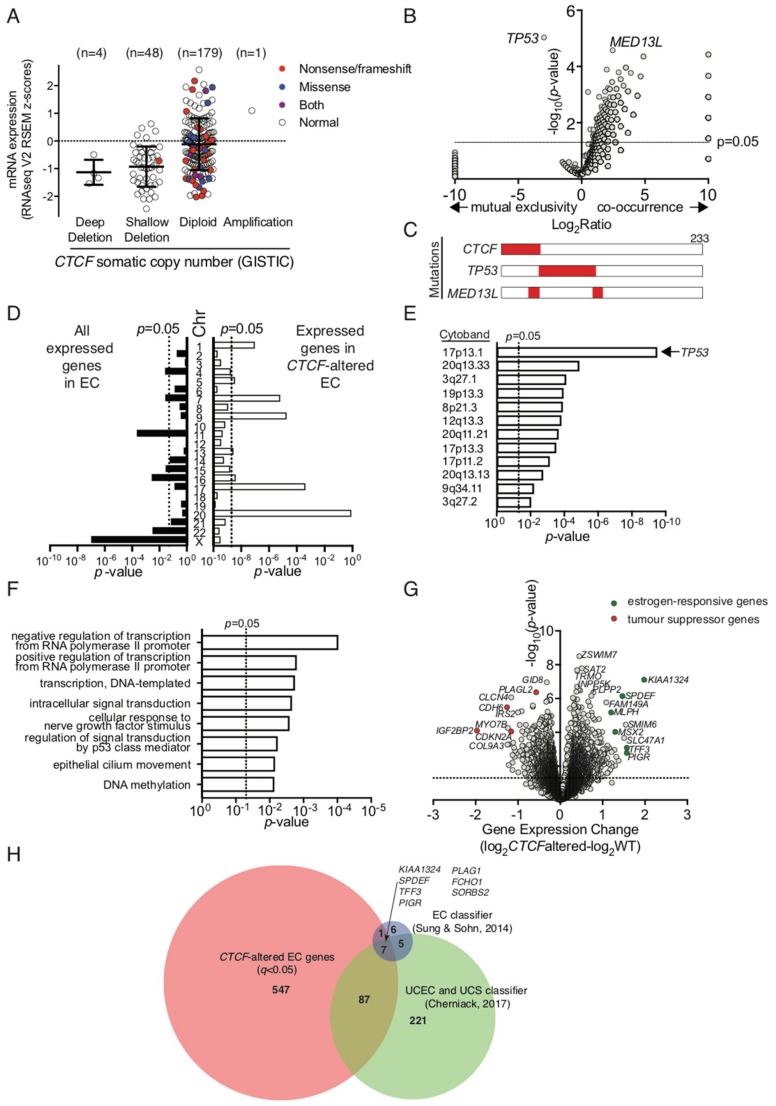
The molecular genetic landscape of *CTCF*-altered endometrial cancers. Gene expression and DNA sequencing data was analysed from the TCGA endometrial carcinoma patient cohort [28]. (**A**) GISTIC analysis of mRNA expression indicative of somatic copy number alterations in endometrial carcinomas. Filled symbols indicate cancers with *CTCF* coding region mutations. (**B**) Plot of significantly co-occurring or mutually exclusive mutant genes with *CTCF*-altered cancer. (**C**) Schematic showing the co-occurrence between *CTCF*, *TP53* and *MED13L* mutations in endometrial cancer (*n* = 233 patient samples). (**D**) Chromosomal distribution of all expressed genes in endometrial carcinoma (*n* = 13,271) vs those differentially expressed (*n* = 642; *q* < 0.05) in *CTCF*-altered cancers (data is normalised for gene density). (**E**) Chromosomal location (cytoband) of genes differentially expressed in *CTCF*-altered endometrial cancers (*q* < 0.05). Data for (**D**) and (**E**) were analysed using the Fisher’s exact test. (**F**) Biological process terms enriched in gene ontology analysis of *CTCF*-altered endometrial cancers (*q* < 0.05). (**G**) Plot of most significantly differentially regulated genes in *CTCF*-altered cancers compared to *CTCF* WT cancers, genes of particular interest are highlighted. (**H**) Genes common to *CTCF*-altered differentially regulated EC genes (642 total; *q* < 0.05) and two gene classifiers used to distinguish uterine corpus endometrial cancers (UCEC) from serous cancers as well as uterine carcinosarcomas (UCS) [39,40].

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
