# Peer review of "CTCF Expression is Essential for Somatic Cell Viability and Protection Against Cancer"

_ijms, 2018, doi:10.3390/ijms19123832_

Round 1

Reviewer 1 Report

Numerous studies showed that CTCF is involved in many cellular processes, including regulation of transcription, chromatin architecture and insulator activity. The MS by Bailey et al.  is dedicated to the study of two different models of CTCF haploinsufficiency in vitro and in vivo. The authors focused on modelling the impact of CTCF haploinsufficiency on the tumour suppressor function of CTCF that has not been previously examined. CTCF depletion in K562 cells by shRNA or by CRISPR/Cas9 system decreased some pro-tumourigenic properties of these cells. However, authors observed the opposite effects in p53 shRNA-immortalized Ctcf+/- MEFs. The authors also analyzed in silico human endothelial carcinoma expression and mutation data and identified locus-specific alterations in gene expression due to CTCF haploinsufficiency. The article is interesting and can be published in the International Journal of Molecular Sciences after a substantial revision.

Major remarks:

1.      MEFs have a set of specific markers, such as Gremlin, Tgfb1, Activin. Are these markers conserved in the initial, immortalized MEFs and after the Ctcf  knockdown?

2.      It would be also helpful if the authors showed the p53 expression levels in the immortalized MEFs.

3.      The supplementary files do not contain any legends.

4.      The authors used in silico analysis to identify locus-specific alterations in gene expression in human endothelial carcinoma cells. It would be helpful to confirm the CTCF-associated changes in gene expression by RT-qPCR.    

Minor remarks:

1.      The quality of the figures in the MS should be improved;

2.     Figure 1A. A double band is present in the tubulin control. It is highly unusual. Did you observe the same signal in cells other than K562?  

Author Response

Numerous studies showed that CTCF is involved in many cellular processes, including regulation of transcription, chromatin architecture and insulator activity. The MS by Bailey et al.  is dedicated to the study of two different models of CTCF haploinsufficiency in vitro and in vivo. The authors focused on modelling the impact of CTCF haploinsufficiency on the tumour suppressor function of CTCF that has not been previously examined. CTCF depletion in K562 cells by shRNA or by CRISPR/Cas9 system decreased some pro-tumourigenic properties of these cells. However, authors observed the opposite effects in p53 shRNA-immortalized Ctcf+/- MEFs. The authors also analyzed in silico human endothelial carcinoma expression and mutation data and identified locus-specific alterations in gene expression due to CTCF haploinsufficiency. The article is interesting and can be published in the International Journal of Molecular Sciences after a substantial revision.

Major remarks:

1.      MEFs have a set of specific markers, such as Gremlin, Tgfb1, Activin. Are these markers conserved in the initial, immortalized MEFs and after the Ctcf  knockdown?

We acknowledge that Gremlin, Tgfb1 and activin are important regulators of embryonic stem cell pluripotency. They are requisite markers in MEFs to sustain the feeder-dependent growth of human embryonic stem cells. However, quantitation of these markers in WT and Ctcf+/- MEFs would not complement this study as their relevance to our assays is uncertain.

2.      It would be also helpful if the authors showed the p53 expression levels in the immortalized MEFs.

The MSCV-based p53.1224 containing retroviral vector (originally described by Dickins et al Nature Genetics 2005) has been widely used for suppressing p53 expression in mouse primary cells and cell lines. ShRNA knockdown of p53 is widely accepted and used as a method of malignant transformation of MEFs. Continuous culture of WT and Ctcf+/- MEFs required in this study would not have been possible without sustained p53 knockdown. Therefore further quantitation of p53 would not add to, or modify, our conclusions.

3.      The supplementary files do not contain any legends.

The supplementary figure legends were provided in the main body of the original submitted paper after the main figures (pages 36-37). The revised Supplementary data file has now been amended with the Supplementary Figure Legends.

4.      The authors used in silico analysis to identify locus-specific alterations in gene expression in human endothelial carcinoma cells. It would be helpful to confirm the CTCF-associated changes in gene expression by RT-qPCR.

Yes, we agree that it would have been helpful to confirm the gene expression signature observed in CTCF-altered cancers by RT-qPCR. However, we do not have access to the samples that were used in the original molecular genetic characterisation of endometrial cancer study (Kandoth et al Nature 2013). 

Minor remarks:

1.      The quality of the figures in the MS should be improved;

All attempts have been made to provide high-quality figures. The uploaded manuscript for review and downloaded Preprint version appeared to be of good quality. It is possible that conversion to the combined pdf for review led to image degradation. No specific comments have been raised about the format, layout and clarity of the submitted figures. However, if the manuscript is accepted, then we are happy to fix any figures at the direction of the editors.

2.     Figure 1A. A double band is present in the tubulin control. It is highly unusual. Did you observe the same signal in cells other than K562?  

We have also observed this double band in HEK293T whole cell lysates.

Reviewer 2 Report

The manuscript by Bailey et al investigated whether CTCF is essential in somatic cells and demonstrate CTCF is a haploinsufficient tumor suppressor gene essential for survival of somatic cells. The study is carefully done and utilize complementary approaches of shRNA KD and CRISPR/cas9. Methods are well described, and results are largely supported by the data. Authors have attempted to correlate their experimental data with the results from TCGA endometrial cancer dataset. However, it would have been better if authors had used endometrial cancer cell line model in this study instead of K562.

Please proofread the manuscript carefully for typos and spelling errors throughout.

Author Response

We have thoroughly reviewed the manuscript for spelling and grammatical errors.

Round 2

Reviewer 1 Report

The authors have addressed most of the concerns of this reviewer.